behaviour, biomechanics

insect flight, passive stability, behavioural strategy, flight control, motion capture, dragonfly

**Author for correspondence:**
Huai-Ti Lin
e-mail: h.lin@imperial.ac.uk

# Dragondrop: a novel passive mechanism for aerial righting in the dragonfly

Samuel T. Fabian, Rui Zhou and Huai-Ti Lin

Department of Bioengineering, Imperial College London, London SW7 2AZ, UK

(iD) STF, 0000-0002-0366-7236; RZ, 0000-0001-9687-0825; H-TL, 0000-0003-3370-244X

Dragonflies perform dramatic aerial manoeuvres when chasing targets but glide for periods during cruising flights. This makes dragonflies a great system to explore the role of passive stabilizing mechanisms that do not compromise manoeuvrability. We challenged dragonflies by dropping them from selected inverted attitudes and collected 6-degrees-of-freedom aerial recovery kinematics via custom motion capture techniques. From these kinematic data, we performed rigid-body inverse dynamics to reconstruct the forces and torques involved in righting behaviour. We found that inverted dragonflies typically recover themselves with the shortest rotation from the initial body inclination. Additionally, they exhibited a strong tendency to pitch-up with their head leading out of the manoeuvre, despite the lower moment of inertia in the roll axis. Surprisingly, anaesthetized dragonflies could also complete aerial righting reliably. Such passive righting disappeared in recently dead dragonflies but could be partially recovered by waxing their wings to the anaesthetised posture. Our kinematics data, inverse dynamics model and wind-tunnel experiments suggest that the dragonfly's long abdomen and wing posture generate a rotational tendency and passive attitude recovery mechanism during falling. This work demonstrates an aerodynamically stable body configuration in a flying insect and raises new questions in sensorimotor control for small flying systems.

## 1. Background

Maintaining body orientation, or attitude control, is an important aspect of animal locomotion, which ensures the delivery of effective forces to the world [1,2]. Maintaining attitude in the air is particularly challenging, as the body can easily freely rotate. To date, most aerial righting studies have focused on flightless animals from cats and other vertebrates [3] to nymphal stick insects [4] and ants [5] that fall from branches [6]. Without a large aerofoil, these animals must rely on the exchange of inertia between body parts, or on the aerodynamic torque of outstretched limbs [3]. Flying animals, however, must master aerial righting as part of the attitude control. Some do this routinely for landing and taking off from a ceiling [7,8], while others do it to recover from an undesirable state [9–11]. Within flightless animals and in hoverflies so far tested, the animal almost always rotates around their longitudinal-axis [8]. This makes mechanical sense as the moment-of-inertia is minimum along the body-axis. Does this general 'roll-out' rule apply to all flapping flight animals or do some choose a particular manoeuvre according to the initial conditions? How much do passive dynamics play a role in aerial righting? These are key questions we address in this study to discover any passive stabilizing mechanism in an agile system, using the dragonfly as a model.

Dragonflies are one of the most successful fliers in nature and their ancestors were the first animals to achieve flight approximately 350 Ma [12]. All species are aerial predators as adults and conduct much of their behaviour while airborne. During prey interception, the dragonfly approaches the prey [13] from

Proc. R. Soc. B 288: 20202676

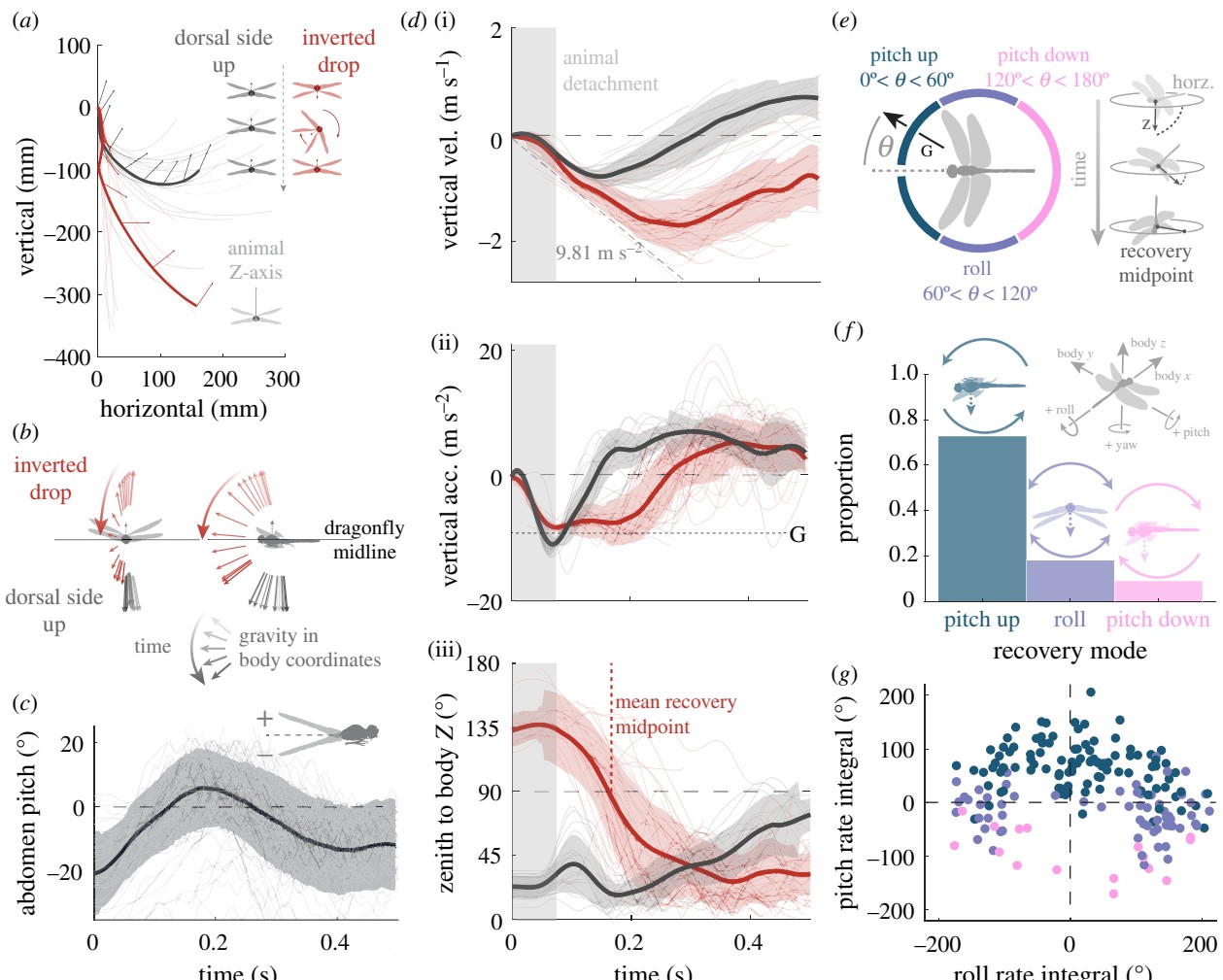

**Figure 1.** Summary figures of inverted drop against dorsal-side-up drop. (*a*) Trajectory plots for dorsal-side-up and inverted drops. Bold lines follow an example trajectory from both populations, with arrows representing the animal's body *z*-vector. Colour schematic for the release inclinations are shown top right, dorsal-side-up in grey, inverted in red. (*b*) Mean positions of the gravitational axis are drawn in the dragonfly's body coordinates at 12 ms intervals. (*c*) The pitch angle of the dragonfly's abdomen is traced in the body coordinates. Mean line is shown in lilac with a shaded region demarking ± s.d. (*d*) The vertical velocity (i), vertical acceleration (ii) and angle from the animal's body *z*-vector to the global zenith (iii) of the falling dragonflies. Bold lines represent means, shaded regions represent ± s.d. (*e*) Recovery mode classification was performed at the recovery midpoint (demonstrated right) when the animal's dorsal vector crossed the horizon. If the measured azimuthal angle of gravity relative to the dragonfly's *x*-axis, *θ*, was less than 60° the animal was predominantly pitching-up, if 60° < *θ* < 120°, the animal was rolling, and if *θ* > 120°, then the animal was pitching-down. (*f*) The relative proportion of each recovery mode if displayed for inverted drops. (*g*) The integrals of the animals' roll- and pitch-rates during the recovery phase (up to 300 ms) are plotted against each other. Colours correspond to the recovery mode classification.

below, terminating the capture episode in an inverted state. What if we initialize the dragonfly from an inverted state? Self-righting behaviours must be performed reliably in flying animals and any passive recovering mechanism could be beneficial. During cruising flight, dragonflies perform both gliding and flapping flight. The ability to glide [14,15] suggests that the airframe may have some self-stabilizing mechanism. We therefore hypothesize that the dragonfly has an aerodynamically stable skydiving pose which facilitates aerial righting given supplemental active control.

## 2. Results

### (a) Modes of active aerial righting

We recorded the three-dimensional kinematics of falling dragonflies using a customized motion capture technique derived from [16,17] (figure 1*a–c*, electronic supplementary material, Methods, and figures S1 and S2). To provide a

benchmark for reaction time, we started by dropping the dragonfly dorsal-side-up using a custom retractable platform (electronic supplementary material, Methods, and figure S1). The dragonfly initialized wing flapping at 48 ± 8 ms (*n* = 8) after the release. This fast reaction enabled the dragonfly to begin accelerating upwards at 161 ± 41 ms (*n* = 20) and reversed its vertical velocity at 305 ± 57 ms (*n* = 20) from release (figure 1*d*). While gaining flight control in mid-air, the dragonfly pitched-up quickly and subsequently damped out the pitch-rate (electronic supplementary material, figure S3). The process took roughly 400 ms. By contrast, the average yaw- and roll-rates followed no such pattern and never exceed 600° s⁻¹ (s.d. from zero) throughout the recovery (electronic supplementary material, figure S3).

We then fully inverted the dragonflies and dropped them from a magnetic platform (figure 1*a–c*, see the electronic supplementary material, Methods). Inverted animals' trajectories had a greater vertical component (figure 1*d*). We defined the time when the animal's dorsal-axis (body-Z)

crosses the horizon as the recovery midpoint. All fully inverted dragonflies were able to right themselves and took on average $198 \pm 48$ ms to cross the recovery midpoint. This increased the time to accelerate upwards to $251 \pm 62$ ms ($n = 39$) (figure 1$d$). Inverted animals achieved a maximal fall velocity of $1.77 \pm 0.41$ m s$^{-1}$ ($n = 43$), more than double that of dorsal-side-up drops at $0.84 \pm 0.24$ m s$^{-1}$ ($n = 20$). Most inverted animals did not fully arrest the downward velocity within the motion capture volume, but those that did took $506 \pm 210$ ms ($n = 7$).

To classify the recovery manoeuvres, we projected the gravity vector onto the dragonfly's egocentric azimuth plane at the time of recovery midpoint (figure 1$e$); this angle will be referred to as 'midpoint $\theta$' throughout this paper. We divided the projected azimuth into three modes: (i) $0°$–$60°$ pitch-up mode: 71%; (ii) $60°$–$120°$ roll mode (left or right): 20%, and (iii) $120°$–$180°$ pitch-down mode: 9% (figure 1$f$). We validated this classification by integrating the roll- and pitch-rates over the critical manoeuvring time window (50–250 ms), as shown in figure 1$g$. We found that while our assigned modes generally matched the integrals, there was significant overlap, suggesting righting is a compound rotation with varying contribution from roll and pitch. The predominance of pitching-up manoeuvre was supported by the mean path of gravity vector in the animals' body coordinates. It travelled from the dorsum to the anterior via the head, showing that the dragonfly's head led the rotation downwards (figure 1$b$).

Interestingly, we did not observe the initiation of flapping (i.e. first wing movement) from the dragonfly until $110 \pm 17$ ms ($n = 10$) into the freefall despite the short sensorimotor latency demonstrated by the dorsal-side-up drops (i.e. 48 ms). The abdomen of the dragonfly went through a stereotypical movement while dropping as seen in figure 1$f$. Its relative angle to the thoracic-axis started approximately $-20°$ pitch (away from the dorsum) and flicked to $+10°$ during the aerial recovery and back to approximately $-10°$ at the end of the manoeuvre. High-frequency oscillations (approx. 40 Hz, the dragonfly's wingbeat frequency) of $2$–$3°$ can be seen in the abdomen after 100 ms, indicating an abdominal movement coordinating with wing flapping.

## (b) The behaviour strategy of aerial righting

To test whether the initial body attitude determines the dragonfly's recovering mode, we inclined the platform by $45°$ with the inverted dragonfly in one of four different inclinations (figure 2$a$–$c$): (i) head-up, (ii) abdomen-up, (iii) left-up and (iv) right-up. Trials were pseudorandomized to minimize any history effects. We used the same recovery mode classification criteria (figure 2$b$). Abdomen-up oriented dragonflies ($n = 29$) had a similar manoeuvre profile to those totally inverted, with 86% of trials demonstrating a pitch-up mode and only 7% a roll or pitch-down modes combined. By contrast, head-up animals ($n = 26$) displayed a choice bifurcation, with pitching-up in only 42% of cases and pitching-down at 47% (rolling manoeuvres accounting for 11%). Side-up (left and right combined $n = 51$) animals predominantly (63%) rolled out of the fall, with 33% pitching-up and 4% pitching-down. The rolling direction of side-up animals was in accordance with their initial inclination, with animals taking the shortest angular path to the vertical (electronic supplementary material, figure S3).

Dragonflies with biased initial attitude took a shorter time ($210 \pm 52$ ms) to reach upward acceleration than fully inverted individuals. Consequently, they also achieved a slower maximum fall velocity of $1.48 \pm 0.51$ m s$^{-1}$ ($n = 108$) (figure 2$c$). We took a snapshot of the animal's attitude at 300 ms to obtain an impression of the animal's attitude for later comparison with other drop types. The animals' roll was $27 \pm 29°$ ($n = 141$) and pitch was $28 \pm 29°$ ($n = 141$). At the end of the active attitude recovery, the animal was typically pitched-down from the horizon.

When we grouped the trajectories by their recovery modes and plotted their mean pitch, yaw and roll (figure 2$d$), the variability of dragonfly righting behaviour was clear. Pitching-rates varied accordingly with recovery modes, with pitch-up mode predictably having the greatest positive pitch-rates ($1005 \pm 305°$ s$^{-1}$), pitch-down mode the greatest negative pitch-rates ($-1623 \pm 242°$ s$^{-1}$). Similarly, roll mode exhibited the greatest maximum roll-rates ($1819 \pm 431°$ s$^{-1}$) and rolling earlier in the trajectory ($170 \pm 63$ ms). Unexpectedly, pitch-down mode also showed greater yaw-rates (max $996 \pm 385°$ s$^{-1}$) than either roll ($238 \pm 242°$ s$^{-1}$) or pitch-up ($213 \pm 85°$ s$^{-1}$) mode. The initial attitude was a stronger predictor of the resulting recovery mode than the animal's initial rotation rates (figure 2$e$). Furthermore, we found that the choice of recovery mode impacted the dragonfly's ability to arrest its fall. A linear fit between the maximum fall velocity and the midpoint $\theta$ (figure 2$f$, $R^2 = 0.53$) demonstrated that the pitch-up manoeuvre led to the fastest fall velocities and pitch-down led to minimal fall velocities.

## (c) The passive dynamics of aerial righting

To isolate the passive mechanism of the aerial righting behaviour, we performed the same dropping experiments on anaesthetized dragonflies. Chilling the dragonflies on the ice for 20 min rendered them unresponsive without any reflexes. The animals were close to freefall acceleration up until $2$ m s$^{-1}$ after which point their acceleration began to slow. Consequently, they also achieved greater fall velocities than when the animals were active ($2.37 \pm 0.19$ m s$^{-1}$) (figure 2$g$). However, anaesthetized dragonflies still righted themselves from an inverted drop with a high success rate (figure 2$h$ and electronic supplementary material, figure S4), taking $178 \pm 105$ ms to reach the manoeuvre midpoint. Anaesthetized animals predominantly pitched-up (78%), with a minority (20%) rolling out, and a single example pitching-down (2%). The angular velocities showed the pitch-up dominance with also high roll component in the behaviour (electronic supplementary material, figure S4). Anaesthetized animals at 300 ms had a mean roll angle of $27 \pm 35°$ ($n = 62$) and a pitch angle of $46 \pm 23°$ ($n = 62$) (electronic supplementary material, figure S4). This pitch angle is slightly more than the measurements from active animals. The initial attitude of the dragonfly was no longer a strong predictor of righting mode, with a clear dominance of pitch-up mode (electronic supplementary material, figure S4).

The observation of passive aerial righting is consistent with our hypothesis that the dragonfly's airframe possesses a passive recovery mechanism. To test the contribution of the wing posture, we first repeated the inverted drops with recently dead dragonflies with flexible joints. Dead dragonflies lost their ability to recover attitude except momentarily transiting through the proper attitude in the process of

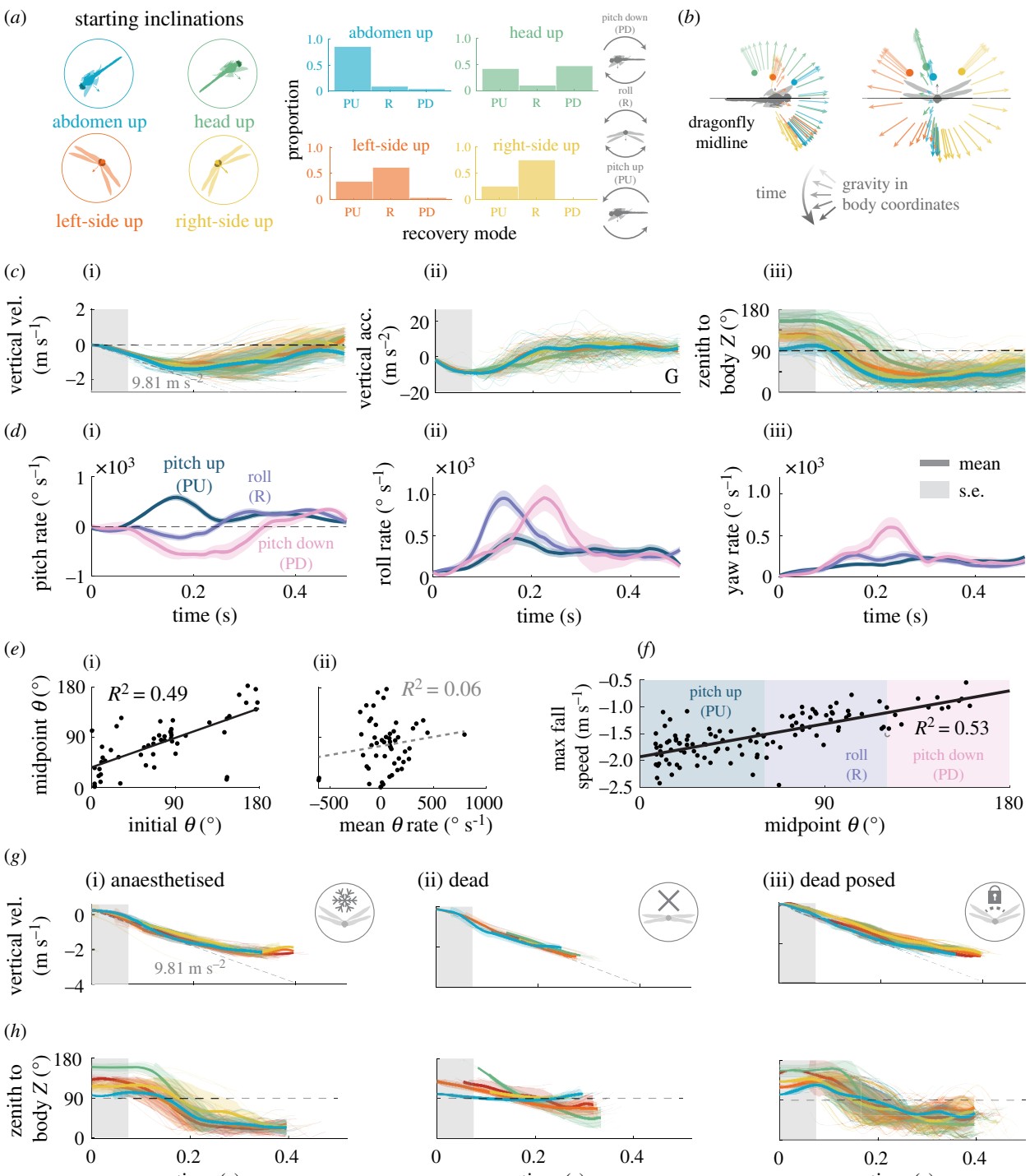

**Figure 2.** Starting body attitude and resultant recovery modes. (*a*) (i) Colour schematic for alternative inclinations applied to the animals' release. (ii) The proportions of different recovering modes for different starting attitude. (*b*) Mean positions of the gravitational axis are traced in the dragonfly's body coordinates. Points represent the mean starting positions. Colours correspond to the initial starting inclinations. (*c*) vertical velocity (i), vertical acceleration (ii) and angle from the animal's dorsal vector to the global zenith of the falling dragonflies (iii). Bold lines represent means, shaded regions represent ± s.d. (*d*) The mean egocentric pitch-rate (i), roll-rate (ii) and yaw-rate (iii) are displayed for the three recovery modes. Shaded regions correspond to ± s.e. (*e*) The initial egocentric azimuthal angle of gravity at start of the drop is plotted against its value at the manoeuvre midpoint (i). The midpoint egocentric azimuthal angle of gravity is plotted against its time derivative (ii). (*f*) The maximum vertical speed plotted against the midpoint $\theta$. Data from the three different passive conditions are presented in comparative columns at the bottom of the figure. Column (i) shows data from dragonflies that have sat on ice for 20 min, (ii) shows data from animals that have recently (less than 24 h) died and (iii) shows data from dead dragonflies that have had their wings posed. (*g*) The vertical velocities of the dragonflies across the three passive conditions. Mean lines for each starting inclination are colour coordinated (see figures 1 and 2). (*h*) Shows the time progression of the angle from the animals' dorsal vectors (body $Z$-vector) to the global zenith. Means of each inclination are shown in bold, and shaded regions represent ± s.d.

tumbling. The mean trajectories show a partial righting (figure 2*g*–*h*) but lacked the righting characteristics seen in anaesthetized animals. The pitch- and roll-rates of dead animals lacked the characteristic pattern and peaks observed in

anaesthetized animals (electronic supplementary material, figure S4). When we took a 300 ms snapshot of the animals' global attitude, we found that the mean body angles were $93 \pm 38°$ ($n = 28$) in roll and $44 \pm 38°$ ($n = 28$) in pitch.

In addition to the lack of control in yaw, the dead animals were not able to reach the level in the roll angle (electronic supplementary material, figure S4).

Reference videos revealed that dead dragonfly wings were not able to take aerodynamic loads owing to the loss of wing joint stiffness. By contrast, anaesthetized dragonflies maintained certain wing dihedrals during the aerial descent (electronic supplementary material, video S1). We measured these wing angles from all four wings using small (less than 1 mm) discs of retro-reflective tape on the leading edge of the wing surface. We found the mean wing-to-wing angles to be $110 \pm 28°$ for forewings, and $140 \pm 17°$ for hindwings. To test the effect of these wing angles, we used beeswax to fix the wing joints of dead dragonflies at different angles to test for how wing poses impacted recovery. For simplicity, we will focus on the condition where we posed the dead dragonfly's wings to the nominal wing angles as in the anaesthetized state. Four other wing poses were explored and discussed in the electronic supplementary material, figure S5.

Posing the wing angles of dead dragonflies rescued a notable proportion of passive aerial righting performance. Most posed animals righted for each of the starting inclinations (figure 2g–h, electronic supplementary material, figure S4b), and overall, they took $190 \pm 52$ ms to reach the manoeuvre midpoint, comparable with anaesthetized animals. The manoeuvre classification was similar to that of anaesthetized animals, with 80% pitch-up and 20% roll mode. The pattern of pitch- and roll-rates was generally recovered (electronic supplementary material, figure S4). The mean 300 ms snapshot attitude of posed animals had a global roll of $52 \pm 47°$ ($n = 140$) and had a pitch of $61 \pm 18°$ ($n = 140$) (electronic supplementary material, figure S4). However, the posed animals frequently entered a spin after recovery midpoint with a yaw-rate of $1332 \pm 791°\,\mathrm{s}^{-1}$ ($n = 140$), compared to $408 \pm 288°\,\mathrm{s}^{-1}$ ($n = 62$) for anaesthetized animals and $294 \pm 284°\,\mathrm{s}^{-1}$ for active animals.

To further isolate the contribution of a wing from the body geometry in the passive righting mechanism, we repeated the inverted drops to recently dead dragonflies with wings removed (electronic supplementary material, figure S4). These wingless dead dragonflies consistently followed a free-fall trajectory throughout the motion capture volume. The body maintains the tendency to pitch-up into a head-first dive but at lower pitch-rate than dragonflies with wings, taking a longer $270 \pm 88$ ms to reach the recovery midpoint. This demonstrates that the body form installs a pitch-up tendency but it is amplified with the appropriated wing poses (electronic supplementary material, figure S4).

## (d) Forces and torques responsible for aerial righting

To understand the dynamics of both active and passive aerial righting, we performed inverse dynamics on the dragonfly body. The model reconstructed the gravity-subtracted forces and torques at the centre of mass before and after the recovery midpoint (electronic supplementary material. figure S6). For active animals before the recovery midpoint, the dragonfly produced aerodynamic forces in a wide range of directions with an emphasis in the ventral-frontal direction (figure 3a). After the recovery midpoint, the dragonfly concentrated the aerodynamic forces in the dorsal direction 81.2° in elevation to counteract the fall. Peak forces reached over 2.5 bodyweight

($\sim$5.69 mN) to arrest the fall and initiate the climb. In the anaesthetized condition, the dragonfly exhibited a somewhat similar pattern but without strong aerodynamic lift in either pre- or post-recovery midpoint period. For posed dragonflies, the aerodynamics forces had a strong sideways component before the recovery midpoint. After the recovery midpoint, the aerodynamic force vector shifts posteriorly compared to the active and anaesthetized conditions. Both anaesthetized and dead posed dragonflies approached their terminal velocity beyond $2\,\mathrm{m\,s}^{-1}$, with the peak aerodynamic force approaching one bodyweight ($\sim$2.28 mN).

The aerodynamic torques in three body-axes are independent in the active dragonfly (figure 3b). A yaw-roll torque coupling emerged in the posed dragonflies after the recovery midpoint. This is a sign of the spiral mode characteristic of fixed-wing aircrafts. The effect weakened in anaesthetized dragonflies where the wing joints compliance existed. This is also reflected in the better yaw stability for the anaesthetized condition. Finally, we found that pitch-down recovery resulted in the minimum loss of altitude with a greater cost in mechanical energy consumption, while the pitch-up recovery resulted in the greatest loss of potential energy with a lesser cost in mechanical energy (figure 3c).

To determine the passive aerodynamic torques as a function of airflow direction, we extracted timeframes in the inverse dynamic model at selected inclinations relative to the velocity vector (figure 3d). As expected, there was a strong righting torque for both the anaesthetized and posed dragonflies in the roll axis. The longitudinal dihedral configuration also led to a converging pitch attitude in the vertical dive. The posed dragonflies exhibited large variation in torque. To verify this mechanism, we subjected posed dragonflies to a vertical wind tunnel (electronic supplementary material, Methods and figure S7) and directly measured the single-axis body torque. The torque pattern was consistent with the model reconstruction in both the roll and pitch axes. Furthermore, recently dead dragonfly lost the toque pattern as the wings gave way to the incoming airflow (figure 3d).

## 3. Discussion

### (a) The behavioural strategy for aerial righting

Most flying systems are limited in the directions they can produce aerodynamic forces. Thus, maintaining an attitude in which lift forces counteract gravity and facilitate forward motion is a cornerstone of flight. How an animal corrects its attitude is closely linked to manoeuvrability and energy efficiency. A good test for the speed and method of attitude correction is initiating a righting reflex by releasing the animal from a static inverted state. This process starts with the detection of freefall. During inverted drops, the dragonfly took twice the time (110 versus 48 ms minimum) to initiate wing movements compared to the dorsal-side-up drops. Given that all the sensory conditions were the same including the removal of feet contact, it is conceivable that such delay in wing movement was part of the manoeuvre. This could be done to delay flapping counter-torque [18], which will slow down the initial body rotation in the first half of the recovery behaviour. The inverse dynamic model showed a ramp-up of body torque starting at $37.8 \pm 9$ ms ($n = 8$), suggesting that some torques were generated before flapping occurred.

Proc. R. Soc. B 288: 20202676

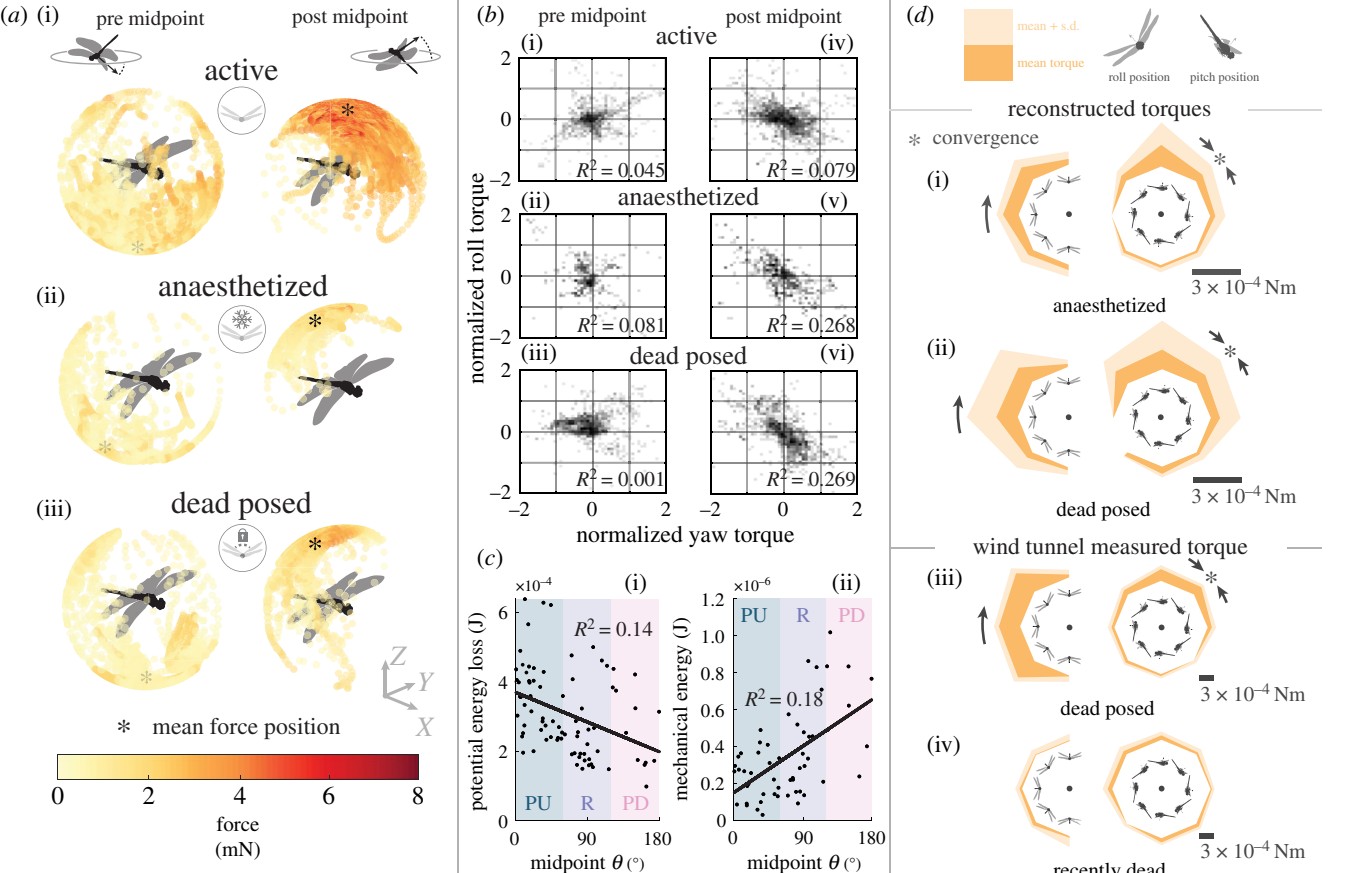

**Figure 3.** Passive aerial attitude recovery and dynamics. (*a*) Direction of force vector and magnitude, before and after the recovery midpoint, for active (i), anaesthetized (ii), and posed (iii) conditions; the average force vector projection is marked by an asteroid. (*b*) Correlation between yaw and roll torques, during pre-midpoint phase (i, ii, iii) and post-midpoint phases (iv, v, vi), for active, anaesthetized and posed conditions. (*c*) (i) The loss of gravitational potential energy and (ii) the minimum mechanical energy are plotted against the egocentric azimuth of gravity at the manoeuvre midpoint. (*d*) The measured torque from inverse dynamics (i, ii) and wind tunnel (iii, iv) are given as a radial distance from the origin, with angle corresponding to animal inclination in either roll (left), or pitch (right). Shaded radial distance shows mean + s.d. Arrows and asteroid mark the converging stable attitude.

Because the correlation between recovery mode preference and the initial body attitude was not present in any passive drop experiment, we concluded that the preference may be initiated by the animal perhaps through feedforward control. The animal could have prepared for a specific manoeuvre based on its starting conditions prior to being released.

Unlike hoverflies [8], dragonflies do not predominantly roll out of inverted drops, instead they generally pitch-up to recover the body attitude. Why does the dragonfly have an inherent preference to perform a pitch-up recovery? Our passive dynamic analyses have demonstrated a strong passive mechanism to do so. The mechanical work required to perform such manoeuvre is also the lowest. However, the pitch-up recovery also incurs the maximum loss of potential energy. Generally, dragonflies falling up-side-down took at least twice the time to arrest their fall velocity than animals falling dorsal-side-up. We found that the roll or pitch-down modes arrested the fall velocity faster and minimized the net loss of potential energy. While the dragonfly's passive tendencies may be to pitch-up, their behavioural variability demonstrates active control.

## (b) A passive mechanism for attitude recovery

Dihedral (inclining the wings upward from horizontal) is a classic approach to install passive stability in fixed-wing aircrafts [19]. While most aircraft dihedral angles are less than

10°, we see some very large dihedral angles in gliding animals: $15 \sim 20°$ for the pipistrelle bat [20] and approximately 45° for pigeons [21]. Dihedral between wings is also present in gliding animals that have little recourse for active control, such as flying fish [22]. The dragonfly passive dihedral angles are approximately 20° and approximately 35° for fore and hindwings, respectively. These angles should provide very high restoring moments in both roll and yaw axes at the cost of higher drag [21]. Why then, were the posed dead dragonflies not able to fully recapitulate the aerial righting performance of the anaesthetised dragonflies? The main difference between the two lies in the passive wing compliance and the wing joint impedance in the anaesthetized dragonflies. The wings of dragonflies passively twist during flapping motion [23], and the wing base joints have elastic compliance. These two forms of compliance may be disrupted by our wing fixation procedure. Under aerodynamic loading in a freefall, the wing dihedral is set by the stiffness of the wing joint. Owing to the viscoelastic nature of muscles, the wing joints of anaesthetized dragonflies may absorb disturbances, similar to that which has been proposed in birds [24]. Finally, the forewing-high/hindwing-low configuration is the default state for the dragonfly gliding flight [14,15], which may confer further passive stability through a longitudinal dihedral [22]. High dihedral may be merely an extension of gliding state to a skydiving state, with varying

degree of passive stability. It is interesting to note that a wingless dead dragonfly also has a tendency to pitch into a dive. This can be explained by the centre of mass being much closer to the head than to the tip of the abdomen. The aerodynamic moment on the abdomen is probably larger than that on the head, leading to a slight torque to put the animal into a dive slowly. Furthermore, it is clear that the appropriated wing posture increases this pitch-up tendency and further installed stability in the yaw and roll axes. However, when the wings are pliant as in recently dead dragonflies, the pitch stability is lost.

Hunting dragonflies often terminate their prey interception at an inverted state with their head higher than the abdomen. They then typically recover the body attitude with a combination of pitch-up and roll. The initial condition is somewhat different as the dragonfly would be carrying a large momentum upwards at this moment. It is unlikely for the freefall dependent passive mechanism to kick in immediately. However, this aerial righting scenario would make an interesting future study in the context of flight behaviour mode transition. Several passive stabilizing mechanisms exist in flapping flight, including flapping counter-torque [9,18] and vibrational stability [25], and further mechanisms have been suggested [26]. These mechanisms resist perturbation and allow active control to correct at a slower time course. It would be interesting in future studies to look at the interaction between the static stability of the dragonfly form and potential dynamic stabilizing mechanisms as mentioned.

## (c) Active control and its constraints

Our inverse dynamic model reveals that the dragonfly can direct its aerodynamic forces in a wide range of angles, but primarily cluster at −93° elevation from the body x-axis during the inverted state. During the active aerial righting behaviour, aerodynamic forces were generated to counteract the gravity even when the dragonfly was in the inverted condition. This suggested that the dragonfly can generate some lift in the inverted state to slow down the descent while performing a recovery body rotation. However, the strongest aerodynamic forces were generated only when the animal is dorsal-side-up.

How much did the inertial control play a role in righting? For simplicity, we excluded the abdomen movement and the inertia of the legs for our current model. This is because wing posture and initial conditions explained most of the biomechanics of aerial righting. However, we noted a stereotypic abdominal movement and that all six legs were held out throughout the righting behaviour. Future work should incorporate these two elements to isolate the contribution of inertial control. These active movements may smooth out the motion of aerial manoeuvres or alter stability as found in hovering hawkmoths [27]. Pitching motion in falling dragonflies appears to be partially a consequence of their long abdomen, as work in manipulated fruit flies suggests it might [26]. Future work should investigate the dynamics of this body part as it may affect the flapping flight not only in inertia [28], but in aerodynamics.

## (d) The contribution from the sensory systems

It is entirely possible that the aerial attitude recovery behaviour is dominated by feedforward control triggered by some mechanosensory signal such as the loss of leg contact. In this study, we deliberately kept the leg contact condition the same for inverted or dorsal-side-up drops. The dragonfly lost leg contact during the release in the same manner. However, such a precise behaviour should require some feedback control to at least arrest the body rotation at the 'correct' attitude. Attitude control is often mediated by the vision and inertial sensation. In many insects, the ocellar system provides a simple horizon detector and drives a dorsal-light-response [29], whereas the compound eyes detect self-motion via optic flow [29]. Together with an inertial sensory system such as the halteres [30], the optical and mechanosensory system provide an insect with the reference orientation to aim for [29]. The dragonfly has a strong ocellar reflex [31] and a highly developed visual system [32]. Therefore, we expect the optical pathway to play an important role in state estimation. Furthermore, while the dragonfly does not have halteres, its head has been proposed to function as an inertial sensor [33]. More recently, hundreds of mechanosensors have been identified on the dragonfly wings (J. Fabian, I. Siwanowicz, M. Uhrhan, M. Maeda, R. J. Bomphrey and H-T. Lin 2020, unpublished data) which could provide aerodynamic and inertial data to supplement the state estimation from vision. What are the contributions from the optical and mechanosensory pathways to the dragonfly aerial righting behaviour? This is a question we shall address in future studies.

In summary, the dragonfly's aerial righting behaviour strategy can be described as conducting the minimum recovery rotation given the initial attitude, with an inherent tendency for the pitch-up recovery. The fall detection can be achieved within 48 ms but the dragonfly typically delays its wing movement in an inverted drop until half-way into the recovery. The tendency for the pitch-up recovery is primarily owing to the wing posture and long abdomen as demonstrated by a variety of passive drops and wind-tunnel measurements. Such recovery relies on the differential drag on the dragonfly's airframe and the active dragonfly sometimes allows such passive recovery at the cost of recovery time. We do not know whether the recovery mode decision process involved energetic consideration and whether the recovery is dominated by feedforward control; future work will investigate this. However, this study thoroughly documented and dissected the dragonfly's pitch-based aerial recovery and its underlying passive mechanism, both of which are novel and somewhat counterintuitive. Our biomechanics and modelling work demonstrate that even at the insect scale, morphology and posture can provide passive aerial stability and reduce the necessity of active control. This lesson from biology can inspire design principles for failsafe attitude recovery in micro aerial systems.

Data accessibility. Data are available from the Dryad Digital Repository: https://doi.org/10.5061/dryad.34tmpg4j7 [34].

Authors' contributions. Experimental design was done by H.T.L., S.T.F. and R.Z. Experiments were performed by S.T.F. and R.Z. Data were analysed by S.T.F., R.Z. and H.T.L. This manuscript was developed by H.T.L. and S.T.F.

Competing interests. We declare we have no conflicting interests.

Funding. European Research Council (ERC StG no. 804315 to H.T.L.).

Acknowledgements. Firstly, we would like to thank our funding body, the European Research Council (ERC StG No. 804315 to H.T.L.), for facilitating our research. We would like to thank Sean Lim for his assistance with laboratory instrumentation. We would like to thank Prof. Richard Bomphrey, Dr Masateru Maeda, Dr Alexandra Yarger and Myriam Uhrhan for their feedback on our work. We would also like to thank Igor Siwanowicz for supplying the CAD model adapted for our simulations.

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
