## [Reviewer comments · Proceedings of the Royal Society B: Biological Sciences]

Review History

RSPB-2020-2676.R0 (Original submission)

Review form: Reviewer 1

Recommendation

Major revision is needed (please make suggestions in comments)

Scientific importance: Is the manuscript an original and important contribution to its field?

Excellent

General interest: Is the paper of sufficient general interest?

Excellent

Quality of the paper: Is the overall quality of the paper suitable?

Good

Is the length of the paper justified?

Yes

Should the paper be seen by a specialist statistical reviewer?

No

Do you have any concerns about statistical analyses in this paper? If so, please specify them explicitly in your report.

No

It is a condition of publication that authors make their supporting data, code and materials available - either as supplementary material or hosted in an external repository. Please rate, if applicable, the supporting data on the following criteria.

Is it accessible?

N/A

Is it clear?

N/A

Is it adequate?

N/A

Do you have any ethical concerns with this paper?

No

Comments to the Author

This is an excellent work proposed by Samuel et al. about the study of passive dynamics involved in the righting of a free falling dragonfly. Authors have shown that passive righting disappears in dead dragonfly but it can be partially recovered by waxing the wings in a particular posture (wing dihedral).

Model based on dynamic inverse is also used (but not described) to analyse the trajectories recorded and the force produced during the righting.

Even if the work is quite impressive, I have the following main concerns to be fully convinced by the aerodynamics effect of the fixed wings:

- additional experiments must be achieved with wingless dragonflies (freshly dead) to clearly separate the contribution of the passive rotation of the body from the wing action. The body is also submitted to drag and lift force with a center of mass not aligned with the center of pressure that might make the body passively rotate.

- a complete description of the inverse dynamic model must be provided as supplementary information including a definition of all the parameters used (Inertia, Mass, friction...) and their numerical values.

- other video footages showing the active righting and the dorsal up fall must be provided.

Minor :

- in figure 1B authors must clearly define what are the angles pitch, roll and yaw and their orientation (negative or positive).

- page 9 line 223 : authors claim that passive mechanism is related to a very low mechanical work. Why not plotting the mechanical work during the manoeuvre to show that it is very low?

- Page 10 line 288 : explain with more details the fabrication of the markers, which kind of press was used?...

- Page 10 line 307 : give a blueprint of the tunnel and give the model of the fan. How was the airflow measured? (Model of the flow meter...) Authors could provide a picture of the setup (tunnel) in the SI.

Review form: Reviewer 2

Recommendation

Major revision is needed (please make suggestions in comments)

Scientific importance: Is the manuscript an original and important contribution to its field?

Marginal

General interest: Is the paper of sufficient general interest?

Acceptable

Quality of the paper: Is the overall quality of the paper suitable?

Acceptable

Is the length of the paper justified?

Yes

Should the paper be seen by a specialist statistical reviewer?

No

Do you have any concerns about statistical analyses in this paper? If so, please specify them explicitly in your report.

No

It is a condition of publication that authors make their supporting data, code and materials available - either as supplementary material or hosted in an external repository. Please rate, if applicable, the supporting data on the following criteria.

Is it accessible?

Yes

Is it clear?

Yes

Is it adequate?

Yes

Do you have any ethical concerns with this paper?

No

Comments to the Author

Summary:

This paper aims to describe the righting dynamics of the dragonfly. It provides a detailed descriptive account of this behavior, but the discussion relating passive and active maneuvers as introduced in the beginning of the paper was lacking.

Comments:

Overall I found it difficult to understand what the conclusions were, and why they were significant. My general impression from the paper is that (live) dragonflies generally right themselves in a manner that minimizes energy use by taking advantage of a wing pose that makes upright (gliding) flight passively stable. If this is the case, then I think the wording and figure layout could be edited to emphasize this. After reading the paper, I am still left wondering if there are cases where live dragonflies perform a different kind of righting maneuver than

anesthetized ones, and if they do, how, when, and why do they use a different maneuver?

I think the aforementioned confusion arises because In the abstract, the paper is introduced in the context of discussing tradeoffs in maneuverability and stability, and in the title the paper references both passive dynamics and control. I found it difficult to understand what the conclusion was with respect to how dragonflies' righting behavior fits into this tradeoff. I would expect to see something like, "under these conditions, dragonflies perform passively stable but slow maneuvers, whereas under these other conditions, dragonflies perform active and fast maneuvers". Instead, it seems that dragon flies mostly perform passively stable maneuvers. Perhaps the point is that although dragonflies are capable of highly dynamic active control maneuvers, as exhibited during prey capture, when they are tasked with righting themselves they employ energetically cheap passive maneuvers?

If the goal of the paper is to establish that dragonflies use passive maneuvers to right themselves, the title and abstract should be edited to make this clearer.

If the goal of the paper is to establish that dragonflies choose whether to use passive vs. active maneuvers to right themselves depending on the circumstances, then additional experiments and analysis are needed. For example, perhaps when dragonflies are dropped in such a way that a passive righting maneuver would cause them to hit the ground they perform a different more agile maneuver to right themselves more quickly in exchange for a higher energy cost?

It does not seem surprising that dragon flies do not roll as much as hover flies to right themselves, since their wings are much longer.

The figure panels are introduced in a very haphazard way. It is customary to introduce figure panels in the text in the order in which they are labelled. E.g. Fig. 1f should be introduced after Fig. 1a-e are introduced and discussed. Same should be true for supplemental figures.

I am very confused by the colors, labels, and diagrams for pitch up and pitch down maneuvers in Fig. 1H-I. I suspect one or more things are wrong. H suggests pitch down is teal colored. But panel I labels teal as pitch up, but the diagram I would describe as pitch down. The text implies that the pitch up for panel I is correct. What is going on here?

In the introduction and discussion the authors indicate that attitude control is of critical importance – some references would be helpful.

Is there a reference for gliding flight in dragonflies?

How does the righting behavior compare to prey capture? Do dragon flies also exhibit primarily changes in pitch during prey capture?

Decision letter (RSPB-2020-2676.R0)

19-Nov-2020

Dear Dr Lin:

Your manuscript has now been peer reviewed and the reviews have been assessed by an Associate Editor. The reviewers' comments (not including confidential comments to the Editor) and the comments from the Associate Editor are included at the end of this email for your reference. As you will see, the reviewers and the Editors have raised some concerns with your manuscript and we would like to invite you to revise your manuscript to address them.

Research ethics:

Use of animals and field studies:

It is a condition of publication that you make available the data and research materials supporting the results in the article. Please see our Data Sharing Policies (<https://royalsociety.org/journals/authors/author-guidelines/#data>). Datasets should be deposited in an appropriate publicly available repository and details of the associated accession number, link or DOI to the datasets must be included in the Data Accessibility section of the article (<https://royalsociety.org/journals/ethics-policies/data-sharing-mining/>). Reference(s) to datasets should also be included in the reference list of the article with DOIs (where available).

Please submit a copy of your revised paper within three weeks. If we do not hear from you within this time your manuscript will be rejected. If you are unable to meet this deadline please let us know as soon as possible, as we may be able to grant a short extension.

Best wishes,
Dr Locke Rowe
mailto:proceedingsb@royalsociety.org

Associate Editor
Comments to Author:

Your paper on the passive dynamics of righting maneuvers in the dragonfly has been reviewed by two experts. Both of the reviewers are positive about the overall question and approach but raise several concerns for the authors to address. In particular, the reviewers suggest that the writing and figure presentation should be improved to make the specific focus questions and take-home story clearer, to allow the reader to more easily follow the specific experimental conditions and the conclusions that can be drawn about the passive and active mechanisms used by dragonflies based on the experimental results.

Reviewer 1 suggests that additional experimental data may be needed to support some of the authors' conclusions. The reviewer raises the concern that passive dynamic contributions can result from both wings and body. If you already have or could readily collect the requested data without substantial delay to the manuscript, addressing this point may strengthen the story of the paper. However, if collection of additional data is not feasible within a short timeframe, I'd suggest that the authors could instead address this point through careful text revisions to clearly state that in some cases the 'passive' condition includes contribution from both the body and wings, and make the best reasonable interpretation from the current data available. Please also ensure that the discussion is appropriately revised to clearly state the limitations of interpretation and any future experiments that would be required to address the remaining questions.

Please submit a revised version that addresses the authors' points, including a point-by-point response to the reviewers. Please thoroughly address all points about the clarity of the questions, interpretation, and significance. Thank you very much for your efforts in contributing to our special issue on 'Stability and maneuverability in animal movement: lessons from biology, modeling, and robotics.' We look forward to receiving your revision.

Reviewer(s)' Comments to Author:

Referee: 1

Comments to the Author(s)

This is an excellent work proposed by Samuel et al. about the study of passive dynamics involved in the righting of a free falling dragonfly. Authors have shown that passive righting disappears in dead dragonfly but it can be partially recovered by waxing the wings in a particular posture (wing dihedral).

Model based on dynamic inverse is also used (but not described) to analyse the trajectories recorded and the force produced during the righting.

Even if the work is quite impressive, I have the following main concerns to be fully convinced by the aerodynamics effect of the fixed wings:

- additional experiments must be achieved with wingless dragonflies (freshly dead) to clearly separate the contribution of the passive rotation of the body from the wing action. The body is also submitted to drag and lift force with a center of mass not aligned with the center of pressure that might make the body passively rotate.

- a complete description of the inverse dynamic model must be provided as supplementary information including a definition of all the parameters used (Inertia, Mass, friction...) and their numerical values.

- other video footages showing the active righting and the dorsal up fall must be provided.

Minor :

- in figure 1B authors must clearly define what are the angles pitch, roll and yaw and their orientation (negative or positive).

- page 9 line 223 : authors claim that passive mechanism is related to a very low mechanical work.

Why not plotting the mechanical work during the manoeuvre to show that it is very low?

Page 10 line 288 : explain with more details the fabrication of the markers, which kind of press was used?...

Page 10 line 307 : give a blueprint of the tunnel and give the model of the fan. How was the airflow measured? (Model of the flow meter...) Authors could provide a picture of the setup (tunnel) in the SI.

Referee: 2

Comments to the Author(s)

Summary:

This paper aims to describe the righting dynamics of the dragonfly. It provides a detailed descriptive account of this behavior, but the discussion relating passive and active maneuvers as introduced in the beginning of the paper was lacking.

Comments:

Overall I found it difficult to understand what the conclusions were, and why they were significant. My general impression from the paper is that (live) dragonflies generally right themselves in a manner that minimizes energy use by taking advantage of a wing pose that makes upright (gliding) flight passively stable. If this is the case, then I think the wording and figure layout could be edited to emphasize this. After reading the paper, I am still left wondering if there are cases where live dragonflies perform a different kind of righting maneuver than anesthetized ones, and if they do, how, when, and why do they use a different maneuver?

I think the aforementioned confusion arises because in the abstract, the paper is introduced in the context of discussing tradeoffs in maneuverability and stability, and in the title the paper references both passive dynamics and control. I found it difficult to understand what the

conclusion was with respect to how dragonflies' righting behavior fits into this tradeoff. I would expect to see something like, "under these conditions, dragonflies perform passively stable but slow maneuvers, whereas under these other conditions, dragonflies perform active and fast maneuvers". Instead, it seems that dragon flies mostly perform passively stable maneuvers. Perhaps the point is that although dragonflies are capable of highly dynamic active control maneuvers, as exhibited during prey capture, when they are tasked with righting themselves they employ energetically cheap passive maneuvers?

If the goal of the paper is to establish that dragonflies use passive maneuvers to right themselves, the title and abstract should be edited to make this clearer.

If the goal of the paper is to establish that dragonflies choose whether to use passive vs. active maneuvers to right themselves depending on the circumstances, then additional experiments and analysis are needed. For example, perhaps when dragonflies are dropped in such a way that a passive righting maneuver would cause them to hit the ground they perform a different more agile maneuver to right themselves more quickly in exchange for a higher energy cost?

It does not seem surprising that dragon flies do not roll as much as hover flies to right themselves, since their wings are much longer.

The figure panels are introduced in a very haphazard way. It is customary to introduce figure panels in the text in the order in which they are labelled. E.g. Fig. 1f should be introduced after Fig. 1a-e are introduced and discussed. Same should be true for supplemental figures.

I am very confused by the colors, labels, and diagrams for pitch up and pitch down maneuvers in Fig. 1H-I. I suspect one or more things are wrong. H suggests pitch down is teal colored. But panel I labels teal as pitch up, but the diagram I would describe as pitch down. The text implies that the pitch up for panel I is correct. What is going on here?

In the introduction and discussion the authors indicate that attitude control is of critical importance – some references would be helpful.

Is there a reference for gliding flight in dragonflies?

How does the righting behavior compare to prey capture? Do dragon flies also exhibit primarily changes in pitch during prey capture?

Author's Response to Decision Letter for (RSPB-2020-2676.R0)

See Appendix A.

Decision letter (RSPB-2020-2676.R1)

02-Jan-2021

Dear Dr Lin

I am pleased to inform you that your manuscript RSPB-2020-2676.R1 entitled "DragonDrop: A novel passive mechanism for aerial righting in the dragonfly" has been accepted for publication in Proceedings B.

The referee(s) have recommended publication, but also suggest some minor revisions to your manuscript. Therefore, I invite you to respond to the referee(s)' comments and revise your manuscript. Because the schedule for publication is very tight, it is a condition of publication that you submit the revised version of your manuscript within 7 days. If you do not think you will be able to meet this date please let us know.

- DNA sequences: Genbank accessions F234391-F234402

- Phylogenetic data: TreeBASE accession number S9123
- Final DNA sequence assembly uploaded as online supplemental material
- Climate data and MaxEnt input files: Dryad doi:10.5521/dryad.12311

Sincerely,

Dr Locke Rowe
Editor, Proceedings B
<mailto:proceedingsb@royalsociety.org>

Associate Editor:
Board Member
Comments to Author:

Thank you for thoroughly addressing the reviewer comments to the earlier draft of your paper, and in particular, for your efforts to add the additional data and methods detail requested. The revised text more clearly and concisely conveys the main findings, and the additional experimental data and methods detail. I am happy to accept your paper for publication in our theme issue, subject to a few additional minor revisions, below.

First, considering the broad audience of Proc B, it could be helpful to add a few sentences at the end of the Discussion on the potential broader implications, particularly in consideration of the interdisciplinary nature of the theme issue. Similarly, you might consider adding a broader concluding statement at the end of the abstract, if space allows.

The figures remain quite complex, and I encourage you to consider the following:

- 1) Are all panels in each figure are really essential to the main findings & story?
- 2) Are all of the separate color schemes really essential? I think I count 6 different color schemes in use in the main figures, including the force gradient. This can be confusing.

Please consider reducing the number of color codes being used where labels could suffice.

In particular, labels would appear to be sufficient for the distinction between 'pre-midpoint' and 'post-midpoint' and for the different torque depictions in Fig 3D.

- 2) Can the figures be visually streamlined for clarity by removing any redundant labels?

For example, by making sure that the time axes are scaled identically between rows whenever possible, and then removing redundant time axis labels (such as Fig. 2 Panel C and G), or, even providing a single time scale bar for all time series plots within the figure, allowing removal of the repeated time axis labels This would allow you to extend the vertical scaling within the figure space, making the trajectory details to be easier to read.

3) In the supplemental materials, it is unclear why the supplemental figures have a black background. This wastes ink for any printed copies, and I'd suggest you consider a more traditional color scheme for the final version. In some of these figures, such as Supplemental Fig 4, you could reduce clutter by removing redundant x-axis labels, as noted similarly for Fig 2 in the main text.

Author's Response to Decision Letter for (RSPB-2020-2676.R1)

See Appendix B.

Decision letter (RSPB-2020-2676.R2)

11-Jan-2021

Dear Dr Lin

I am pleased to inform you that your manuscript entitled "DragonDrop: A novel passive mechanism for aerial righting in the dragonfly" has been accepted for publication in Proceedings B.

Your article has been estimated as being 9 pages long. Our Production Office will be able to confirm the exact length at proof stage.

Open Access

Paper charges

All supplementary materials accompanying an accepted article will be treated as in their final form. They will be published alongside the paper on the journal website and posted on the online

figshare repository. Files on figshare will be made available approximately one week before the accompanying article so that the supplementary material can be attributed a unique DOI.

Sincerely,
Proceedings B
<mailto:proceedingsb@royalsociety.org>

Appendix A

Response to Referees

DragonDrop: A novel passive mechanism for aerial righting in the dragonfly

Samuel T. Fabian, Rui Zhou, and Huai-Ti Lin

DOI 10.1098/rspb.2020.2676

Response to Associate Editor

Your paper on the passive dynamics of righting maneuvers in the dragonfly has been reviewed by two experts. Both of the reviewers are positive about the overall question and approach but raise several concerns for the authors to address. In particular, the reviewers suggest that the writing and figure presentation should be improved to make the specific focus questions and take-home story clearer, to allow the reader to more easily follow the specific experimental conditions and the conclusions that can be drawn about the passive and active mechanisms used by dragonflies based on the experimental results.

We thank the reviewers for their feedback on our manuscript. They have raised salient and important considerations and we hope that they find our revisions make a marked improvement. We have addressed the reviewer comments in turn below and have tracked the changes in the main text document attached at the end of this response. Where possible, we have streamlined the original text to make our main messages more succinct to the reader.

Reviewer 1 suggests that additional experimental data may be needed to support some of the authors' conclusions. The reviewer raises the concern that passive dynamic contributions can result from both wings and body. If you already have or could readily collect the requested data without substantial delay to the manuscript, addressing this point may strengthen the story of the paper. However, if collection of additional data is not feasible within a short timeframe, I'd suggest that the authors could instead address this point through careful text revisions to clearly state that in some cases the 'passive' condition includes contribution from both the body and wings, and make the best reasonable interpretation from the current data available. Please also ensure that the discussion is appropriately revised to clearly state the limitations of interpretation and any future experiments that would be required to address the remaining questions.

We managed to perform additional experiments as requested by the Reviewer 1 and the results are now included in the discussion. We have also pointed out the limitations of our interpretation for the results and suggested future investigations.

Please submit a revised version that addresses the authors' points, including a point-by-point response to the reviewers. Please thoroughly address all points about the clarity of the questions, interpretation, and significance. Thank you very much for your efforts in contributing to our special issue on 'Stability and maneuverability in animal movement: lessons from biology, modeling, and robotics.' We look forward to receiving your revision.

A tracked-change version of the resubmitted manuscript will appear that editing occurred everywhere. This is mainly due to the effort to reduce word counts and to condense the information. We have also moved the majority of our methods to a supplemental file to allow us to expand and give full details as requested.

Response to Referee 1:

This is an excellent work proposed by Samuel et al. about the study of passive dynamics involved in the righting of a free falling dragonfly. Authors have shown that passive righting disappears in dead dragonfly but it can be partially recovered by waxing the wings in a particular posture (wing dihedral). Model based on dynamic inverse is also used (but not described) to analyse the trajectories recorded and the force produced during the righting.

Even if the work is quite impressive, I have the following main concerns to be fully convinced by the aerodynamics effect of the fixed wings:

- additional experiments must be achieved with wingless dragonflies (freshly dead) to clearly separate the contribution of the passive rotation of the body from the wing action. The body is also submitted to drag and lift force with a center of mass not aligned with the center of pressure that might make the body passively rotate.

In accordance with your suggestion, we have conducted and analysed new wingless dragonfly drops ($n = 80$). There were indeed from freshly dead dragonflies. We have added the results to the main body of the text (line 151 - 157) and to Figure S4. Briefly, you are quite correct that the falling body generates turning forces independently of the wings. The most characteristic of which is a positive pitching motion (pitching up), demonstrating that the body's centre of mass is ahead of its centre of pressure (toward the head). However, the effect is considerably smaller than the effect from the wings. Nevertheless, the additional experiments do suggest that the dragonfly's long abdomen influences the righting manoeuvre and we have added our interpretation to the discussion (line 231-236).

- a complete description of the inverse dynamic model must be provided as supplementary information including a definition of all the parameters used (Inertia, Mass, friction...) and their numerical values.

A full explanation of the inverse dynamic model is now included in our supplemental methods. We have also added a new figure (Figure S6), which should give all the required details of the model to enable reproduction.

- other video footages showing the active righting and the dorsal up fall must be provided. We have made a more complete supplementary video with new, higher quality video. All experimental styles should now be covered within the supplementary video.

Minor:

- in figure 1B authors must clearly define what are the angles pitch, roll and yaw and their orientation (negative or positive).

We have added arrows explicitly stating the pitch, roll, and yaw axes to Figure 1f.

- page 9 line 223 : authors claim that passive mechanism is related to a very low mechanical work. Why not plotting the mechanical work during the manoeuvre to show that it is very low?

We have added a new panel to Figure 3c that gives the mechanical work required for our measured flights.

Page 10 line 288 : explain with more details the fabrication of the markers, which kind of press was used?...

Further detail has been added to the methods section to better describe the marker manufacture process. We have also added a panel to our methods figure that illustrates how markers were made.

Page 10 line 307 : give a blueprint of the tunnel and give the model of the fan. How was the airflow measured? (Model of the flow meter...) Authors could provide a picture of the setup (tunnel) in the SI. We have added a more complete drawing of our wind tunnel to a new supplementary figure S7. We have also added details of the anemometer used to gauge windspeed to the supplemental methods.

Response to Referee 2:

Overall I found it difficult to understand what the conclusions were, and why they were significant. My general impression from the paper is that (live) dragonflies generally right themselves in a manner that minimizes energy use by taking advantage of a wing pose that makes upright (gliding) flight passively stable. If this is the case, then I think the wording and figure layout could be edited to emphasize this. After reading the paper, I am still left wondering if there are cases where live dragonflies perform a different kind of righting maneuver than anesthetized ones, and if they do, how, when, and why do they use a different maneuver?

We now understand the confusion and revised the introduction to mention only questions that we answered in the study. See below for further justifications.

I think the aforementioned confusion arises because In the abstract, the paper is introduced in the context of discussing tradeoffs in maneuverability and stability, and in the title the paper references both passive dynamics and control. I found it difficult to understand what the conclusion was with respect to how dragonflies' righting behavior fits into this tradeoff. I would expect to see something like, "under these conditions, dragonflies perform passively stable but slow maneuvers, whereas under these other conditions, dragonflies perform active and fast maneuvers". Instead, it seems that dragon flies mostly perform passively stable maneuvers. Perhaps the point is that although dragonflies are capable of highly dynamic active control maneuvers, as exhibited during prey capture, when they are tasked with righting themselves they employ energetically cheap passive maneuvers? If the goal of the paper is to establish that dragonflies use passive maneuvers to right themselves, the title and abstract should be edited to make this clearer.

We acknowledge that the initial wording made our hypotheses and conclusions unclear. We have reworded the title, much of the abstract, background, and discussion for clarity and to make take-away messages clearer. To summarise, we found that dragonflies have a passive, pitching-up righting mechanism available to them, determined by their long bodies and wing dihedral. We find that many of the active animals also pitch up out of dives, but a significant proportion do not. The proportion rolling or pitching-down out of dives increases when we biased the initial starting conditions. We do not yet know why exactly animals may use different righting modes, but the presence of a passive mode, and the measured variability in behaviours demonstrate that active animals can input control to change their righting mode. We hope that the new, extensive text modifications leave the reader with these conclusions.

If the goal of the paper is to establish that dragonflies choose whether to use passive vs. active maneuvers to right themselves depending on the circumstances, then additional experiments and analysis are needed. For example, perhaps when dragonflies are dropped in such a way that a passive righting

maneuver would cause them to hit the ground they perform a different more agile maneuver to right themselves more quickly in exchange for a higher energy cost?

We do not propose to suggest the exact reasons why dragonflies opt for alternative righting modes whilst active. To do so, you are quite correct that extensive further experimentation would be required. It is our aim to expand on this question, with perturbations during righting and cruising flight, to further pick apart passive stability and active effects. For now, we highlight that in some active cases, dragonflies right themselves in ways distinct from the passive mode, and that this frequency is increased when their starting condition is biased at particular inclinations.

It does not seem surprising that dragon flies do not roll as much as hover flies to right themselves, since their wings are much longer.

The aspect ratios of dragonflies and many hoverflies are quite similar (wingspan to body length). While the dragonfly has longer wings, they also have significantly longer body. In fact, the slender form of the dragonfly body makes the moment of inertia in the roll axis 6 times smaller than the pitch axis. Making the rotation much easier in the roll axis compared to a fly with more round body shape. Most other studied animals primarily right themselves roll-wise, thus the pitch-righting of dragonflies was somewhat counterintuitive and unique. Of course, the wing loading of the dragonfly is significantly lower than that of a fly. The larger wing area can produce drag to resist body roll motion, but it can equally produce more aerodynamic torques to assist body roll motion. In any case, the dragonfly having larger wings does not intuitively explain why a pitch manoeuvre is preferable, until we see the behaviour.

The figure panels are introduced in a very haphazard way. It is customary to introduce figure panels in the text in the order in which they are labelled. E.g. Fig. 1f should be introduced after Fig. 1a-e are introduced and discussed. Same should be true for supplemental figures.

We have rearranged figures and text so that their flow matches. Figure order should now match the order that they appear in the main text. Thank you for pointing this out.

I am very confused by the colors, labels, and diagrams for pitch up and pitch down maneuvers in Fig. 1H-I. I suspect one or more things are wrong. H suggests pitch down is teal colored. But panel I labels teal as pitch up, but the diagram I would describe as pitch down. The text implies that the pitch up for panel I is correct. What is going on here?

You are quite correct that the colouring was accidentally reversed in the demonstration figure. Thank you for pointing it out and we have corrected the error. In addition, we have gone through the main figures and made some diagrams greyscale whenever the use of colour was not meaningful.

In the introduction and discussion the authors indicate that attitude control is of critical importance – some references would be helpful.

In animal locomotion, attitude control is somewhat intuitively an important factor. However, we have now cited a textbook and a relevant paper we discussed in the manuscript. Thank you for the request.

Is there a reference for gliding flight in dragonflies?

We have cited a reference (line 41) that includes descriptions of dragonfly gliding and another that estimates their wing posture, including wing dihedral.

How does the righting behavior compare to prey capture? Do dragon flies also exhibit primarily changes in pitch during prey capture?

We have added this discussion in the manuscript (line 237 - 240). In short, the attitude recovery after the prey capture has a key difference in its initial condition: it carries a large upward momentum from the acceleration to the target. The recovery therefore appears to have more equal contribution of pitch and roll components. However, we will extend our study in this type of condition in the future to determine whether a non-static initial condition would impact the dragonfly's strategy for aerial righting.

Appendix B

Response to Associate Editor

DragonDrop: A novel passive mechanism for aerial righting in the dragonfly

Samuel T. Fabian, Rui Zhou, and Huai-Ti Lin

DOI 10.1098/rspb.2020.2676

Thank you for thoroughly addressing the reviewer comments to the earlier draft of your paper, and in particular, for your efforts to add the additional data and methods detail requested. The revised text more clearly and concisely conveys the main findings, and the additional experimental data and methods detail. I am happy to accept your paper for publication in our theme issue, subject to a few additional minor revisions, below.

Thank you for accepting our manuscript, we are happy to hear that it has met the review criteria with your approval. We have addressed your editorial points in turn below.

First, considering the broad audience of Proc B, it could be helpful to add a few sentences at the end of the Discussion on the potential broader implications, particularly in consideration of the interdisciplinary nature of the theme issue. Similarly, you might consider adding a broader concluding statement at the end of the abstract, if space allows.

We have adapted the final sentence of the abstract to read “*This work demonstrates an aerodynamically stable body configuration in a flying insect and raises new questions in sensorimotor control for small flying systems*”, which we hope should lend a more general take-away. We have also added a sentence to give a broader implication to our work on line 291: “*Our biomechanics and modelling work demonstrate that even at the insect scale, morphology and posture can provide passive aerial stability and reduce the necessity of active control. This lesson from biology can inspire design principles for failsafe attitude recovery in micro aerial systems*”, we hope that this should give a broader applicability to our research.

The figures remain quite complex, and I encourage you to consider the following:

1) Are all panels in each figure are really essential to the main findings & story?

Throughout the revision process, many panels have been removed or reduced. We think that those that remain cannot easily be removed without damaging the paper’s interpretability or reducing the evidence presented for our arguments. However we have rearranged and simplified panels, where possible.

2) Are all of the separate color schemes really essential? I think I count 6 different color schemes in use in the main figures, including the force gradient. This can be confusing. Please consider reducing the number of color codes being used where labels could suffice. In particular, labels would appear to be sufficient for the distinction between 'pre-midpoint' and 'post-midpoint' and for the different torque depictions in Fig 3D.

We have reduced the colour usage throughout all the figures. Existing colormaps were colour-blind friendly, but have been tweaked to facilitate greater monochromatic (luminance-based) contrast.

2) Can the figures be visually streamlined for clarity by removing any redundant labels?

For example, by making sure that the time axes are scaled identically between rows whenever possible, and then removing redundant time axis labels (such as Fig. 2 Panel C and G), or, even providing a single time scale bar for all time series plots within the figure, allowing removal of the repeated time axis labels This would allow you to extend the vertical scaling within the figure space, making the trajectory details to be easier to read.

We have reduced the axis footprint where possible across all figures. (including supplemental). We have not totally removed the axis line itself in favour of a set time-scale, as we believe the proximity of the ticked line aids the viewer in determining where specific course features are. However, we have removed axis and tick labels wherever possible.

3) In the supplemental materials, it is unclear why the supplemental figures have a black background. This wastes ink for any printed copies, and I'd suggest you consider a more traditional color scheme for the final version. In some of these figures, such as Supplemental Fig 4, you could reduce clutter by removing redundant x-axis labels, as noted similarly for Fig 2 in the main text.

In keeping with your request, all supplemental figures now feature a white background. Within supp. figures 3 & 4 especially, we have removed superfluous text around the axes wherever possible. Colour usage has been reduced throughout the supplemental figures in favour of monochromatic shades.